# Pilot Study: Assessing the Expression of Serum Lactate Dehydrogenase and Peripheral Leukocyte Ratios in Canine Oral Malignant Melanoma

**DOI:** 10.3390/vetsci9080421

**Published:** 2022-08-09

**Authors:** Ben Murray, Kelly L. Bowlt Blacklock

**Affiliations:** Hospital for Small Animals, Royal (Dick) School of Veterinary Studies, University of Edinburgh, Easter Bush Campus, Midlothian EH25 9RG, UK

**Keywords:** canine, melanoma, LDH, leukocyte, metastasis

## Abstract

**Simple Summary:**

This study aimed to assess the expression of novel blood biomarkers in dogs with oral malignant melanoma, an aggressive and common oral cancer in dogs. The results of this pilot study suggest that the concentrations of lactate dehydrogenase, which can be easily measured with blood samples, are significantly higher in dogs with oral malignant melanoma. These blood biomarkers have been shown to be of prognostic value in human melanoma patients, meaning that they can act to predict the potential behaviour of the tumour. Therefore, research into our canine patients should be viewed as being potentially very valuable, as the discovery of easily measurable prognostic biomarkers could significantly further not only our understanding of the underlying physiology of melanoma itself, but also change the way veterinary surgeons investigate and treat the disease in the future. This study lays the foundations for further, more extensive investigation, into the topic.

**Abstract:**

Measurement of blood biomarkers such as lactate dehydrogenase (LDH) and peripheral leukocyte ratios have been shown to be of prognostic value in human melanoma patients. Previous veterinary studies have demonstrated that changes in these values are detectable in multiple canine cancer patients. However, to the authors’ knowledge, no studies have yet demonstrated an increase in LDH in canine oral malignant melanoma patients, nor has the effect of metastasis on LDH levels been explored. This retrospective pilot study included 18 dogs, of which 10 were healthy controls, 5 OMM patients with metastasis and 3 without metastasis. Serum LDH was measured and pre-treatment peripheral leucocyte ratios were calculated. LDH was measurable within all patient groups and a statistically significant difference in LDH levels was detected between patients with OMM and healthy controls (*p* < 0.05); however, no significant difference was detected between patients with or without metastatic disease. This study suggests that serum LDH levels are significantly increased in dogs with OMM compared to healthy controls, paving the way for further research to investigate the prognostic value of this biomarker.

## 1. Introduction

Oral malignant melanoma (OMM) is the most common oral neoplasm in the dog, accounting for 30–40% of all canine oral malignancies [1,2,3]. There is no known sex predilection and it is typically a disease of older dogs, most often diagnosed in Scottish terriers, golden retrievers, chow chows, poodles and dachshunds [4]. The site of tumour growth is most commonly located in the gingiva but can be found anywhere within the oral cavity including the internal lip, tongue and hard palate [1]. OMM is aggressive and locally invasive with variable rates of metastasis reported, ranging from 30.3% to 74% to the local lymph nodes and 14.0% to 92% to the lungs and other organs [3]. Melanomas have an extremely diverse and unreliable spectrum of biological behaviour. The biological behaviour of OMM can be predicted using several clinical, biochemical and histopathological factors including anatomic site, size, clinical stage, mitotic index, Ki67 expression, degree of pigmentation, nuclear atypia and immunohistochemistry [3,4,5,6,7].

The treatment for patients with OMM begins with local disease control, of which en bloc surgical excision alongside mandibular and medial retropharyngeal lymphadenectomy is prioritised in most cases [1,8]. Adjunctive therapies may include radiotherapy, electrochemotherapy, immunotherapy or chemotherapy where appropriate [3]. Many recent veterinary studies have focussed on the immunogenicity of melanoma, with novel adjuvant immunotherapies showing initially encouraging results [9,10]. The median survival time for dogs with OMM treated using surgery alone are 511 to 874 days, 160 to 818 days and 168 to 207 days for stages I–III, respectively, with metastatic disease being the most common cause of death [3].

Recently within human medical oncology, there has been an increased interest in the use of circulating blood biomarkers, such as lactate dehydrogenase (LDH), as prognostic indicators of disease progression and treatment effectiveness [11,12]. LDH catalyses the conversion of pyruvate to lactate during anaerobic respiration in normal and neoplastic cells, with increased serum LDH levels believed to reflect high metabolic activity within the hypoxic and highly active cellular environment of malignant tumours [13,14]. To this end, assessment of LDH has been included in the updated American Joint Committee on Cancer melanoma staging system, with multiple studies having confirmed a correlation between serum LDH levels and decreased survival times in human patients with advanced stage melanoma [15,16,17]. Within Veterinary Medicine, studies have shown significant elevations in serum LDH levels in dogs with lymphoma, mammary gland and oral tumours compared to healthy controls [13,14,18,19].

As well as LDH levels, recent human studies have shown blood neutrophil to lymphocyte ratios (NLR) and lymphocyte to monocyte ratios (LMR) to be of prognostic value in various cancers including melanoma, with an increased NLR and decreased LMR representing negative prognostic indicators [20,21,22]. These haematological changes are theorised to reflect the effect of central inflammation on the development and progression of cancer, with lymphocytes aiding in the production of an immune response against the tumour and monocytes conversely being recruited to produce growth factors and cytokines within the tumour leading to immunosuppression and angiogenesis, creating optimal conditions for growth within the tumour [6]. The prognostic impact of these ratios has also been assessed in several canine malignancies including lymphoma, soft tissue sarcoma, osteosarcoma, mast cell tumours, and OMM. Although results vary between studies, multiple papers have demonstrated these ratios to be of prognostic value in veterinary patients [23,24,25,26].

To this author’s knowledge, no previous studies have assessed serum levels of LDH specifically in patients with OMM, nor have they assessed the effect of metastasis on serum LDH levels. The aim of this pilot study was to compare LDH levels from stored frozen serum in OMM patients against those of healthy controls, as well as to assess whether there was a statistically significant difference in LDH, NLR and LMR values between patients with or without evidence of metastasis.

## 2. Materials and Methods

### 2.1. Patient Selection Criteria

Records from the Hospital for Small Animals, University of Edinburgh were searched for client-owned dogs with a confirmed histological diagnosis of OMM between June 2012 and December 2021. Dogs with confirmed OMM through histopathology and that had undergone full clinical staging to determine the presence of metastasis to the local lymph nodes and lungs were considered eligible for entry into this study. Clinical staging included lymph node assessment through either fine needle aspiration (FNA) or histopathology when available as well as head and thorax CT scan, or alternatively 3-view thoracic radiography. Inclusion criteria included availability of pre-treatment haematology and leukocyte differential counts taken at patient presentation as well as evidence of full clinical staging to assess for the presence of metastasis. Patients were only included in the non-metastatic group if full lymph node histopathology was available to rule out metastasis. Not all patients included in this study had frozen serum samples for assessment of LDH but were included purely for assessment of leukocyte ratios. Exclusion criteria included those patients with any evidence of hepatic or cardiac disease either at presentation or in the previous month that may interfere with LDH values, as well as patients that had received corticosteroids within the past month that may interfere with leukocyte differential counts.

Data retrieved from patient records for each dog included signalment (age, breed and sex), tumour location, evidence of lymph node metastasis at initial staging, evidence of lung metastasis at initial staging, method of staging (FNA vs. histopathology/CT vs. X-ray), date of sample collection and pre-treatment haematology results including leukocyte counts.

Ethical approval for this study was granted through the Veterinary Ethical Review Committee, University of Edinburgh (126.21). Written forms signed by owners prior to recruitment for this study confirmed owners’ consent to data collection as well as storage and use of spare blood from clinical sampling for research purposes.

Control values for LDH were measured from the frozen serum of clinically healthy patients with no evidence of recent or concurrent disease on history or clinical examination as a part of the University of Edinburgh biobank programme.

### 2.2. Evaluation of Serum LDH

After collection in serum tubes, samples were centrifuged at 2500 rpm for 10 min at room temperature. Serum samples were then stored at −80 °C in a clinical biobank within 24 h of collection. For the purpose of this study, frozen samples were defrosted at room temperature on a lab roller mixer and immediately run through Beckman Coulter AU480 analyser (Beckman Coulter Ltd, London, UK) using their reagent, category number OSR6126. Results were displayed in U/L. The normal range for LDH at our laboratory was 21–212 U/L. Given that localised or metastatic disease could be present, total serum LDH was measured, rather than individual isoenzyme values which have been shown to vary based on the affected organ [27].

### 2.3. Evaluation of NLR and LMR

Haematological analysis including leukocyte differentials and complete blood count, was performed on whole blood in ethylenediaminetetraacetic (EDTA) anti-coagulant at the time of patient presentation prior to any treatment. All samples were run either at the Easter Bush Veterinary Pathology Unit using a Siemens Advia 2120 (Siemens Healthcare GmbH, Erlangen, Germany), or in-house using a Procyte Dx machine (IDEXX Laboratories, Inc., Westbrook, ME, USA). As measurements were taken using multiple analysers, standard reference intervals were used when considering leukocyte counts, shown in Table 1. NLR and LMR were calculated using ratios of the absolute count of neutrophil to lymphocytes and the absolute count of lymphocytes to monocytes.

### 2.4. Statistical Analysis

Statistical analysis was performed using Minitab^®^ (version 20.3) with statistical significance set to *p* < 0.05. Analysis was applied to explore the relationship between serum LDH values of control, non-metastatic and metastatic groups, as well as assess for difference in NLR and LMR in non-metastatic and metastatic groups. One way ANOVA testing was used to compare values between groups.

## 3. Results

### 3.1. Patient Population

A total of 15 patients with OMM were considered eligible for this study. Of the 15 patients, 8 had stored frozen serum available for LDH measurement and 14 had pre-treatment haematological data available. One patient did not have pre-treatment haematological data but was included in this study for use in LDH groups only. In addition to this, a total of 10 healthy canine patients were included as a control group for LDH measurement. Patient details are included in Table 2.

The median age of all dogs in the sample population was 10 years (range 5–13 years). 9/15 patients had evidence of metastasis to the local lymph nodes (9/9) or lungs (1/9), whereas 6/15 had no evidence of metastasis. Lymph node metastasis was detected by FNA in 5/9 cases and histopathology in 4/9 cases. All non-metastatic cases had local metastasis ruled out through lymph node histopathology. One dog displayed a mild neutrophilia (13.22 × 10^9^) and two patients displayed a mild lymphopenia (0.52 × 10^9^ and 0.65 × 10^9^).

### 3.2. Serum LDH Values between Groups

Serum LDH was measured in a total of 18 patients, 5 with known metastasis to the local lymph nodes or lungs, 3 without evidence of metastasis and 10 control patients. The median average serum LDH value for the healthy control group was 299 U/L (range 228–1368 U/L) compared to 748 U/L (range 198–2032 U/L) for the OMM group (Figure 1). Within the OMM group, average LDH values were 1612 U/L (range 327–1762 U/L) and 611 U/L (range 198–2032 U/L) for non-metastatic and metastatic groups, respectively. There was a significant difference between all OMM patients and the control group (*p* = 0.049), however there was not a statistically significant difference between OMM patients with or without metastasis (*p* = 0.40) (Figure 2).

### 3.3. NLR and LMR between Groups

NLR and LMR from pre-treatment haematological data were assessed in 14 patients with OMM, 9 with evidence of metastasis and 5 without evidence of metastasis. A healthy control group was not available when assessing this variable, so analysis was focussed on comparison between metastatic and non-metastatic groups. The median NLR and LMR for all OMM patients with or without metastasis were 4.83 (range 1.14–11.57) and 3.83 (range 1.11–7.67), respectively. For patients without metastasis, median NLR and LMR were 4.29 (range 1.14–11.57) and 2.43 (range 1.11–5.25), respectively. For patients with metastasis, median NLR and LMR were 5.37 (range 1.94–9.11) and 4.00 (range 1.5–7.67), respectively. There was not a statistically significant difference between metastatic and non-metastatic groups for NLR (*p* = 0.80) or LMR (*p* = 0.46) (Figure 3 and Figure 4).

## 4. Discussion

The purpose of this retrospective study was to assess whether statistically significant differences in serum LDH could be detected between canine patients with OMM and healthy controls, as well as to assess whether significant differences existed between those patients with and without evidence of metastasis. In addition to this, we aimed to assess the effect of metastasis on NLR and LMR values in canine OMM patients. Previous studies have investigated serum LDH values in dogs with any oral tumour however to this author’s knowledge, no study had focussed on serum LDH values in OMM, nor had they assessed the effect of metastasis on LDH levels [19]. One recent study had assessed NLR and LMR in canine OMM patients and found no statistically significant correlation with any current accepted prognostic indicators including clinical stage [6]. Given the known prognostic value of LDH, NLR and LMR in human malignant melanoma patients, further investigation into their prognostic impact is warranted in our canine patients. The use of these values in veterinary medicine is an appealing idea, as all can be easily and affordably measured from routine blood samples, often taken during clinical staging [11,12,15].

It has been shown that even in well perfused, normoxaemic masses, energy is often obtained through anaerobic rather than aerobic respiration through the so called ‘Warburg effect’. It has therefore been proposed that increased serum LDH in cancer patients acts as a reflection of increased glycolytic activity in neoplastic cells, where LDH acts to drive anaerobic respiration through catalysing the conversion of pyruvate to lactate whilst regenerating NADH to NAD^+^ [28].

Similarly to LDH, the use of peripheral leukocyte ratios such as NLR and LMR has been investigated widely in human medicine with recent studies demonstrating their role as prognostic indicators in a number of human malignancies including malignant melanoma. Studies have shown an increased NLR is often seen in patients with a greater clinical stage and a decrease in LMR associated with shorter survival times [21,22]. The association between systemic inflammation and cancer is well recognised. Although the anti-tumour effects of neutrophils has been clearly demonstrated, relative neutrophilia has been shown to increase the release of a number of inflammatory markers including growth factors, anti-apoptotic markers and pro-angiogenic markers, all aiding tumour growth and progression [20,29]. Lymphocytes also play a key role against cancer in the body, acting through both humoral and cellular anti-tumour immune responses [30]. Low lymphocyte counts are frequently seen in advanced human cancer patients and have been correlated with poorer overall survival [22]. Conversely, monocytes recruited by tumours can act to release growth factors, promote angiogenesis and release immunosuppressive cytokines, all acting to produce a favourable tumour microenvironment [22]. Previous veterinary studies have largely demonstrated promising results when assessing leukocyte ratios as prognostic indicators in various malignancies including lymphoma, mast cell tumours, feline injection site sarcomas and osteosarcomas [23,24,25,26]. One recent study evaluated pre-treatment NLR and LMR in canine OMM patients but found no correlation with any known prognostic factors or indeed survival time, suggesting these ratios may be of little use as prognostic tools in canine OMM [6].

In this current study, there was a statistically significant difference in serum LDH levels between OMM patients and healthy controls. However, no significant difference in serum LDH was detected in OMM patients with or without metastasis. These data suggest that serum LDH could potentially act as a biomarker in canine OMM patients, in agreement with previous studies assessing its use in other canine malignancies. Similarly to LDH, no statistically significant differences were detected in NLR and LMR levels between canine OMM patients with or without evidence of metastasis. This supports the previous study by Camerino et al., which demonstrated no difference in NLR or LMR with increasing clinical stage in OMM patients [6].

This study had several limitations which may have contributed to a type 2 error. Firstly, our sample size was limited, especially with regard to LDH samples, where the availability of stored serum presented a major challenge. Additionally, the use of frozen LDH samples presents the potential for enzyme degradation over time, leading to potentially inaccurate readings. The retrospective nature of this study also presents a major limitation. Therefore, results obtained in this study especially with regard to serum LDH should be approach cautiously.

Given the results in this study, further investigation into the use of prognostic and diagnostic biomarkers such as LDH in canine OMM is worthwhile, with the effect of metastasis and tumour progression being of particular interest going forward. This current study, although limited in scope, acts as a pilot study, clearly demonstrating the relationship between OMM and serum LDH levels and laying the foundations for further research. In the future, prospective studies utilising a larger patient population, full histopathological assessment of lymph nodes and immediate measurement of serum LDH would be needed to assess these biomarkers further.

## Figures and Tables

**Figure 1 vetsci-09-00421-f001:**
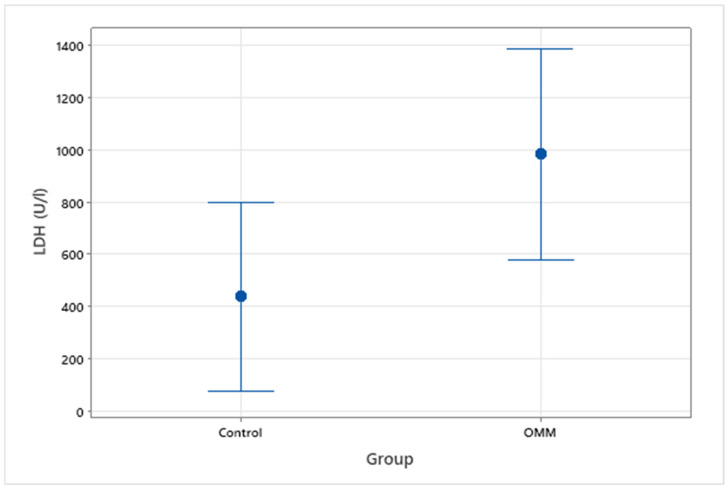
Interval plot to show serum LDH levels (U/L) in clinically healthy dogs vs. OMM patients.

**Figure 2 vetsci-09-00421-f002:**
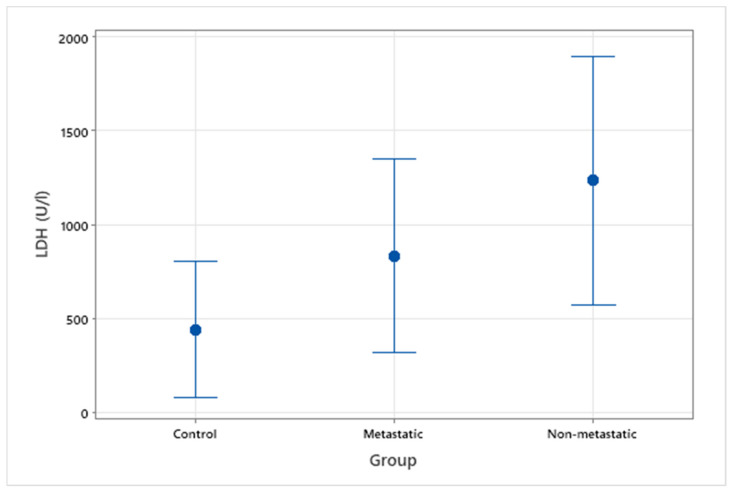
Interval plot of serum LDH levels (U/L) in clinically healthy dogs, patients with OMM with metastasis, and patients with OMM without metastasis.

**Figure 3 vetsci-09-00421-f003:**
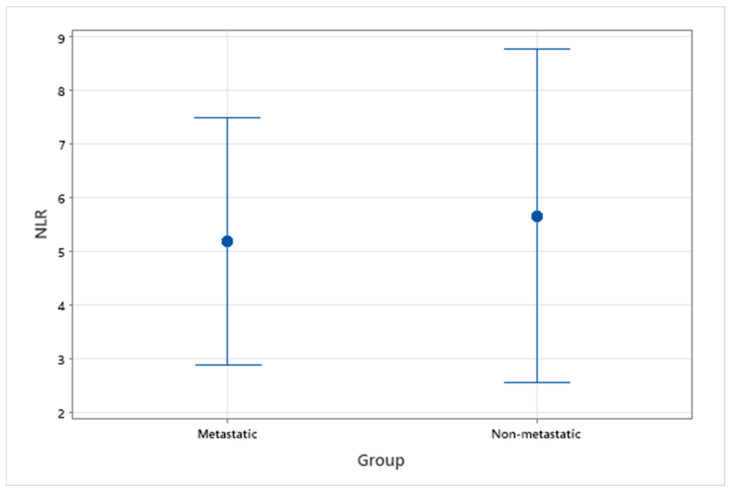
Interval plot of pre-treatment NLR in patients with OMM, with and without evidence of metastasis.

**Figure 4 vetsci-09-00421-f004:**
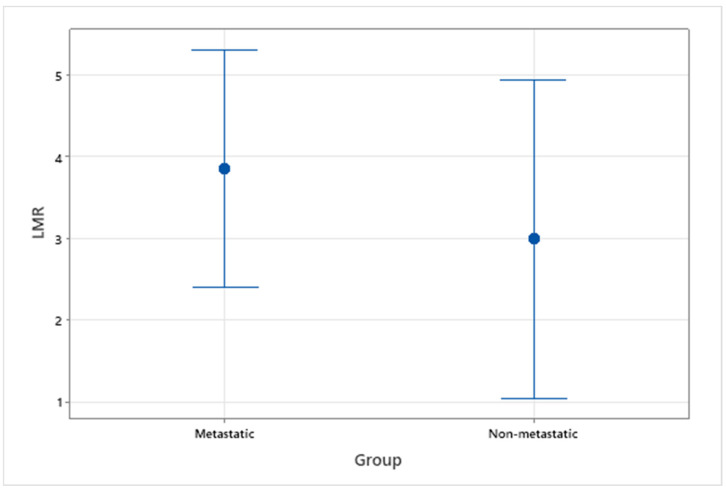
Interval plot of pre-treatment LMR in patients with OMM, with and without evidence of metastasis.

**Table 1 vetsci-09-00421-t001:** Standard canine reference values for haematological parameters.

Haematological Parameter	Reference Values
White blood cell count (×10^9^/L)	6.0–15.0
Neutrophil count (×10^9^/L)	3.6–12.0
Lymphocyte count (×10^9^/L)	0.7–4.8
Monocyte count (×10^9^/L)	0–1.5

**Table 2 vetsci-09-00421-t002:** Patient population characteristics.

Patient	Age (Years)	Breed	Sex	Group	Tumour Location	LN Metastasis	Lung Metastasis	LN Assessment	Chest Assessment	LDH (U/L)	NLR	LMR
**1**	11	Cross breed	FE	Metastatic	Gingival	Yes	No	FNA	3-view chest X-ray	442	2.46	4.00
**2**	12	Labrador	ME	Metastatic	Gingival	Yes	No	FNA	CT	611	7.73	3.67
**3**	9	Labrador	FN	Metastatic	Lingual	Yes	Yes	FNA	CT	198	9.11	1.50
**4**	10	Cocker spaniel	FN	Metastatic	Gingival	Yes	No	FNA	CT	2032	5.37	4.67
**5**	10	Cocker spaniel	FN	Metastatic	Gingival	Yes	No	Histopathology	CT	885	7.50	1.46
**6**	13	Tibetan terrier	ME	Metastatic	Gingival	Yes	No	FNA	CT		1.94	5.33
**7**	8	Golden retriever	ME	Metastatic	Gingival	Yes	No	Histopathology	CT		3.43	4.20
**8**	10	Golden retriever	FN	Metastatic	Gingival	Yes	No	Histopathology	CT		3.04	7.67
**9**	9	Red setter	ME	Metastatic	Gingival	Yes	No	Histopathology	3-view chest X-ray		6.15	2.17
**10**	5	Labrador	MN	Non-metastatic	Gingival	No	No	Histopathology	3-view chest X-ray	1612	3.40	5.00
**11**	8	Labrador	ME	Non-metastatic	Gingival	No	No	Histopathology	3-view chest X-ray	1762		
**12**	11	Labrador	FN	Non-metastatic	Gingival	No	No	Histopathology	CT	327	4.29	2.43
**13**	13	Border terrier	FN	Non-metastatic	Gingival	No	No	Histopathology	CT		7.90	1.11
**14**	12	Bearded collie	MN	Non-metastatic	Gingival	No	No	Histopathology	3-view chest X-ray		11.57	1.17
**15**	12	Cocker spaniel	FN	Non-metastatic	Gingival	No	No	Histopathology	3-view chest X-ray		1.14	5.25
**16**	1	Hungarian vizsla	MN	Control						505		
**17**	10	Cross breed	FE	Control						294		
**18**	11	Labrador	MN	Control						488		
**19**	8	Spaniel (unspecified)	FE	Control						228		
**20**	2	Golden retriever	FN	Control						278		
**21**	4	Goldenretriever	ME	Control						405		
**22**	3	Cocker spaniel	FN	Control						303		
**23**	11	Labrador	FN	Control						1368		
**24**	1	Cocker spaniel	FN	Control						235		
**25**	2	Golden retriever	MN	Control						290		

## Data Availability

The data presented in this study are available on request from the corresponding author.

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
