# Peer review of "Pilot Study: Assessing the Expression of Serum Lactate Dehydrogenase and Peripheral Leukocyte Ratios in Canine Oral Malignant Melanoma"

_vetsci, 2022, doi:10.3390/vetsci9080421_

Round 1

Reviewer 1 Report

The authors chose to measure lactate dehydrogenase (LDH) in patients with canine oral malignant melanoma (OMM). The research includes only 5 verified cases of canine metastases and 3 unconfirmed cases. The sample size is insufficient to draw any conclusions from the experiment. Even though the authors acknowledged the need for more investigation, I would strongly advise that in this case, it is insufficient to support any assertion. Because of this, the research can't be taken seriously without the extra information.

The authors omitted the abstract entirely from their article, which is a minor but crucial formatting note. An abstract was only submitted in the online form of the journal website.

Author Response

Thank you for your comments. We appreciate the time taken to review our manuscript

Reviewer 2 Report

Oral malignant melanoma is the most important oncological problem in the oral cavity of dogs next to squamous cell carcinoma. This study is scientifically and clinically important. The methodology of the study is appropriate. The limitation of this study is the small number of patients whose results were analysed and therefore this study should be considered as a pilot study. As noted by the authors, it would have been better to use also non-frozen samples, nevertheless, meaningful results were obtained indicating the potential use of LDH as a biomarker. Would it be possible to provide information on whether the animals were sterilised or not?

Author Response

  • Alterations have been made both in the introduction and discussion sections of the manuscript to further clarify that this is indeed a pilot study, which aims to act as a springboard for further prospective research.
  • For the reviewers’ reference, sterilisation status of each patient included in the study is reported in table 2: patient population characteristics.

Reviewer 3 Report

The manuscript is well arranged about serum lactate dehydrogenase and peripheral leukocyte ratios in canine oral malignant melanoma. Detailed information on the subject is given and the findings are described in detail. Important information has been given for researchers who will work with this subject in the field of veterinary oncology.

In 19. Reference,  “Thai J Vet Med. 2019. 49(3):283-288.” This information has been given twice. This information after the authors should be removed.

 Choisunirachon N, Klansnoh U, Phoomvuthisarn P, Pisamai S, Thanaboonnipat C, Rungsi A. Thai J Vet Med. 2019. 49(3):283-288.The expression of serum lactate dehydrogenase in canine oral tumors. Thai J Vet Med. 2019. 49(3):283-288

In 28,30,31,32 references, journal names are in italics. These journal names should be written normally.

Author Response

  • These font errors have been corrected in the revised manuscript.

Reviewer 4 Report

The manuscript entitled “Assessing the expression of serum lactate dehydrogenase and peripheral leukocyte ratios in canine oral malignant melanoma” is a research with an interesting subject. However, major limitation can be highlighted. As the own authors stated, there is a very low number of sample enrolled and with a small set of patients with serum sample to evaluate lactate dehydrogenase. Moreover, very limited of association were made on the manuscript. Therefore, see my specific comments below:

1.       The manuscript is lacking the basic structure, such as abstract, authors did not provide an abstract for the manuscript

2.       Also author’s affiliation is lacking.

3.       No further information was assessed and correlated in the manuscript with the different variables. i.e. no histopathological characteristics was assessed, such as degree of pigmentation, mitosis count, etc.

4.       Authors could increase the manuscript statistics by a statistician. Maybe performed correlation analysis for some parameters and also trying a multivariable analysis. A stronger statistic could be important.

5.       The standard deviation is too high in both groups making the impression that lactate dehydrogenase is not a good marker or other important limitations could be interfering in this result (time of sample frenzying).

6.       Since authors did not have a strong results description (or maybe because the results are not strong), the discussion section is superficial.

Author Response

  • An abstract and author affiliation have been added.
  • The use of further variables such as mitotic count, degree of pigmentation etc. were considered during study conception but decided to be outwith the scope of the current study. Furthermore, given our limited patient number, it was considered unlikely that any significant correlation would be found between our biomarkers and these variables, with the proposed additional information only acting to dilute the relevant novel data this research presents.
  • A statistician was consulted regarding the possibility of multi-variable analysis. In their opinion, given the limited sample number, multi-variate analysis was not considered appropriate for this study.
  • It is unsurprising that the S.D will be high with low patient numbers, however the results of this pilot study, in the authors opinion, are significant enough to warrant further research. The ultimate goal of this study is to stimulate interest in blood biomarkers for OMM and encourage further research. We feel this is achieved.
  • The discussion section of this study details at length our current understanding of the relevant biomarkers. Furthermore we carefully critique our own research and make suggestions as to future avenues of study.

Round 2

Reviewer 1 Report

No more comments

Author Response

Thank you for reviewing our paper. I would like to reply to your comments but cannot see any comments attached.

The authors are native English speakers